# Body Composition Estimation in Breeding Ewes Using Live Weight and Body Parameters Utilizing Image Analysis

**DOI:** 10.3390/ani13142391

**Published:** 2023-07-23

**Authors:** Ahmad Shalaldeh, Shannon Page, Patricia Anthony, Stuart Charters, Majeed Safa, Chris Logan

**Affiliations:** 1Faculty of Environment, Society and Design, Lincoln University, Lincoln 7647, New Zealand; shannon.page@lincoln.ac.nz (S.P.); patricia.anthony@lincoln.ac.nz (P.A.); stuart.charters@lincoln.ac.nz (S.C.); 2Faculty of Agribusiness and Commerce, Lincoln University, Lincoln 7647, New Zealand; majeed.safa@lincoln.ac.nz; 3Faculty of Agriculture and Life Sciences, Lincoln University, Lincoln 7647, New Zealand; chris.logan@lincoln.ac.nz

**Keywords:** body composition, body condition score, body parameters, fat, live weight, ewes’ conditions, image analysis

## Abstract

**Simple Summary:**

Monitoring animal condition is integral for maintaining a healthy flock, increasing ewe productivity, refining animal nutrition, and identifying suitable animals for slaughter. Accurate determination of the body compositions (the amount of fat, muscle, and bone) of ewes can be used to evaluate their conditions, which provides key information to make management decisions. Farmers currently rely on live weight (LW) and body condition score (BCS) to evaluate the health statuses of ewes. This research proposed the use of visual imaging to determine body dimensions, which are then used in combination with LW to predict the body compositions of ewes. The results showed a correlation between fat, muscle, and bone weight determined by computerized tomography (CT) and the fat, muscle, and bone weight estimated by the live weight and body parameters calculated using the image processing application. The results showed an optimal fat of 9% of LW for ewes during the production cycle. If the percentage of fat is less than or more than 9%, farmers have to take action to improve the conditions of the animals to ensure the best performance during weaning and ewe and lamb survival during the next lambing.

**Abstract:**

Farmers are continually looking for new, reliable, objective, and non-invasive methods for evaluating the conditions of ewes. Live weight (LW) and body condition score (BCS) are used by farmers as a basis to determine the condition of the animal. Body composition is an important aspect of monitoring animal condition. The body composition is the amount of fat, muscle, and bone; knowing the amount of each is important because the information can be used for better strategic management interventions. Experiments were conducted to establish the relationship between body composition and body parameters at key life stages (weaning and pre-mating), using measurements automatically determined by an image processing application for 88 Coopworth ewes. Computerized tomography technology was used to determine the body composition. Multivariate linear regression (MLR), artificial neural network (ANN), and regression tree (RT) statistical analysis methods were used to develop a relationship between the body parameters and the body composition. A subset of data was used to validate the predicted model. The results showed a correlation between fat, muscle, and bone determined by CT and the fat, muscle, and bone weight estimated by the live weight and body parameters calculated using the image processing application, with r^2^ values of 0.90 for fat, 0.72 for muscle, and 0.50 for bone using ANN. From these results, farmers can utilize these measurements to enhance nutritional and management practices.

## 1. Introduction

Monitoring and improving individual animal performance is one mechanism to lift economic returns for sheep farming operations [1]. Body condition score (BCS) is a quick and easy way to evaluate ewe condition, using a rating value between one and five; one represents poor, and five represents obese. A ewe in good condition will typically have a BCS between 2.5 and 3.5. BCS is most often defined to the nearest 0.5 increments [2,3,4]. BCS can provide an indication of percentage fat by well-trained evaluators; however, it is a subjective measure [5,6,7,8]. Due to the subjectivity of the BCS, the development of a new scale is required to provide a more accurate estimate of fat [8]. 

Body composition is the amount of fat, muscle, and bone; knowing the amount of each is important because the information can be used for better farm strategic management interventions [9,10]. It is particularly important to check the ewe’s condition at weaning and pre-mating to ensure the ewe’s condition recovers after weaning, as ewes must be in an optimal condition at pre-mating [11]. These are particularly crucial times in a ewe’s life cycle to make sure it is ready for mating, ensuring ewe and lamb survival in the next lambing, and ensuring the best performance of the animal during weaning [11,12,13,14]. Furthermore, the body composition profiles of ewes between gestation and pre-mating indicate the animals’ reproductive performance [15].

Medical methods such as ultrasound [16,17,18], DEXA [10] and CT [19] are used as research methods for body composition estimation. The accuracy of CT, compared with dissection for determining body composition, achieved r^2^ values of 0.98 for fat, 0.92 for muscle, and 0.83 for bone [20,21]. However, these methods are time-consuming, expensive, and require expertise, equipment, and special medical procedures [10,18,22].

Body parameters have been used for the determination of yak and ewe LWs using image measurements [23,24,25] and using physical measurements [26,27,28,29]. Body parameters determined by image analysis have been used as a reliable guide for estimating the body size of sheep [30,31,32,33], newborn lamb size [34,35], predicting fat for pigs [36], and carcass characteristics estimation of sheep [37].

Due to BCS subjectivity, the complexity of medical methods and body parameters have not been widely applied to estimate body composition on-farm [38]. Therefore, this paper proposed an alternative method using body parameters determined by an image analysis application coupled with LW to estimate the body compositions of Coopworth ewes and compare this with the BCS, which indicates fat only. To estimate the body composition, three statistical models, MLR, ANN, and RT, were investigated. These models were compared to find the best model between dependent and independent variables in terms of r^2^ value and error percentage.

## 2. Materials and Methods

### 2.1. Experimental Protocol and Approach

Body composition data were determined by CT. LW was measured using a 3-way weigh crate scale manufactured by Prattley. BCS was assessed, and the ewe’s ID was recorded by a farm manager. Ewes were then fixed into a neck brace to take physical body parameters and capture top and side images, and the ewes were then released. Statistical analysis was undertaken to determine which body parameters could predict body composition. Data were collected from 88 ewes aged 2–4 years at Lincoln University farm at weaning and pre-mating. A set of 74 ewes were scanned in both experiments, and 14 ewes were scanned only at weaning. The wool impact factor was determined after the length of the wool was tested and taken into account [39].

### 2.2. Data Collection

#### 2.2.1. CT Scans

Ewes were CT-scanned at the Lincoln University CT lab using a CT750 HD machine manufactured by GE Healthcare. The CT slice measurements were measured by the CT operator using the STAR 6.15 software. 

The animals were fasted with the water removed for 12 h. Lincoln University SOP 83 and Animal Ethical Committee (AEC) #642 approval were followed for the capturing of CT data, LW and physical body measurements, and BCS. Before scanning, the ewes were tranquilized with Acezine 10 mg administered intramuscularly at 0.1 mL/per 10 kg (e.g., a 60 kg ewe was given 0.6 mL, and a 70 kg ewe was given 0.7 mL, and so on) to relax their muscles and keep stress to a minimum. After 20–30 min and once sedated, the ewe was loaded into a wooden CT scanning stretcher in the sternal recumbence position (on its back). Two scout scans were taken, one from the top half of the body and another from the bottom half. 

#### 2.2.2. Image Capturing

The ewes were secured in a neck brace, and then top and side images were taken. The neck brace stopped the ewes from moving and helped with keeping them in a standard standing position. Three top images were taken using a GoPro 7 camera (12 Megapixel) mounted orthogonally, with a height of 2350 mm from the ground. The side camera was a Canon DSLR 750D and was used to capture up to three visual side-view images, with 6000 mm between the center of the ewe and the camera and a 760 mm height between the ground and the camera, as shown in Figure 1. The camera had a 24.2 Megapixel CMOS sensor, a DiG!C 6 image processor, and an EF-S 18–135 mm f/3.5–5.6 IS STM lens. While taking the top and side images, two small whiteboards were used to report the animal ID, which was used later during the analysis to identify the animal. 

The top body parameters were chest width, width, rump width, top length, and top area (top body area without the head), as shown in Figure 2.

The side body parameters were body length from the brisket to the top point of the leg, the side length from the rump dock to the forearm, the angle length from the neck to the top point of the hock, the height from the lowest point of the front hoof to the top shoulder, the depth from the top rack to the lowest point of the belly, and the side area (side body area without head and legs), as shown in Figure 3. Body parameters of width, top length, front height, body length, angle length, depth and side length, chest width, rump width, back height, top, and side areas were determined by an image-processing application at weaning and using pre-mating scans. This application was developed using image-processing functions from the OpenCV library. Once an image is uploaded to the application, it will determine body parameters. These body parameters with LW are then used in the application to estimate the body composition using the prediction models.

### 2.3. Statistical Analysis

The data from the weaning and premating experiments were combined, yielding 162 observations. A set of 138 observations were used as training data, and the remaining 24 observations were used as testing data. For ANN, 114 observations were used as the training set, 24 observations were used as a validation set, and the remaining 24 observations were used as a testing set. Factor analysis was used to check the correlation between the body parameters of the training data. 

## 3. Results

### 3.1. Descriptive Statistics

The descriptive statistics were presented for the training data. The descriptive analysis showed the minimum, maximum, mean, and standard deviations of all ewes, as shown in Table 1. The amount of fat had a wider range, compared with muscle and bone. The minimum fat amount determined was 0.88 kg, compared to a maximum of 17.64 kg, while the minimum bone amount determined was 2.03 kg, and the maximum was 3.77 kg. 

Ewes that had a BCS between 2.5–3.5 had a minimum fat = 1.29 kg and a maximum fat = 13.86 kg, with an average fat = 5.55 kg. Within this BCS range, ewes had a minimum LW = 47.5 kg and a maximum LW = 78.5 kg, with an average LW = 59.2 kg. Then, based on the average BCS and average LW, an optimal fat amount for ewes was found to be around 9% of the LW. 

### 3.2. Application Accuracy

The results showed an average absolute difference of 4% for body parameters measured by the custom ruler and body parameters determined by the image-processing application, as shown in Table 2. The impact of wool length on the body parameters was tested on five ewes to find the adjustment amount. After adjustment, all parameters decreased slightly, and LW decreased by 1200 g.

### 3.3. Factor Analysis

The collinearity check in Table 3 shows the collinearity between independent variables based on three components, where each component had variables that had high collinearity between them. The first component included chest width, angle length, body length, side length, top length, width, rump width, top area, and side area. The second component included the BCS and LW. The third component included height and back height.

### 3.4. Fat

After testing all possible combinations of the independent variables for the estimation of fat based on factor analysis, as shown in Table 4, a relationship between the independent variables’ weights and chest widths with the dependent variable fat was established. The final multivariate regression model estimated the fat to have an r^2^ = 0.79 and an RMSE = 1.34, with no co-linearity obtained. The result was tested using 24 ewes and showed that r^2^ = 0.87, and RMSE = 1.40.

For ANN, the predicted model accounted for fat, with an r^2^ = 0.88 and an RMSE = 1.17 for the training data. This model had two inputs—LW and chest width—with one hidden layer and one output (fat), with no collinearity. However, all variable models were examined to show the possible maximum result, with an r^2^ = 0.95 and RMSE = 1.22, as shown in Table 5. The known amount of fat was tested, and a relationship was found to validate the fat prediction model using the test data, with an r^2^ = 0.94 and an RMSE = 1.01. 

For the regression tree, different combinations between body parameters were analyzed and compared. The model with the highest r^2^ value and lowest RMSE for the prediction of fat used two variables: LW and chest width, with r^2^ = 0.67 and RMSE = 1.75. The model was validated, and the result showed that r^2^ = 0.72, and RMSE = 1.42.

### 3.5. Muscle

The highest r^2^ to estimate muscle was found between the LW and width model. The model was statistically significant (*p*-values ≤ 0.05), with r^2^ = 0.52 and RMSE = 1.03, with no co-linearity obtained, and the equation is displayed in Table 6. The results for the test data showed an r^2^ = 0.41 and an RMSE = 0.86 between the actual and predicted muscles.

One model was selected for muscle prediction, with r^2^ = 0.77 and RMSE = 1.26, using ANN with one hidden layer used and three inputs (LW, rump width, front height). The highest predicted model accounted for muscle, with r^2^ = 0.79 and RMSE = 1.20 for using all variables and r^2^ = 0.72 and RMSE = 1.03 for test data, as shown in Table 7.

The regression tree model for the prediction of muscle from independent variables used LW, width, and chest width. The results showed that muscle had an r^2^ = 0.25 and an RMSE = 1.27 for training data and an r^2^ = 0.21 and an RMSE = 1.19 for test data.

### 3.6. Bone

The highest r^2^ was found for the relationship using all of the variables, but this relationship was rejected, as it was against the factor analysis. The next highest r^2^ value to estimate bone was found for the relationship between LW and width, with an r^2^ = 0.26 and an RMSE = 0.87, as shown in Table 8.

The results of the test data showed an r^2^ of 0.34 and an RMSE = 0.26 between the CT bone and the predicted bone. It was noticed that the bone had a very small variation between 2.03 kg and 3.77 kg, which could explain the low r^2^ value, as mentioned in descriptive statistics.

All combinations of the independent variables were compared according to the highest r^2^ value and the lowest RMSE. A relationship of r^2^ = 0.75 with an RMSE = 2.40 to estimate bone was found using all variables. However, one model had three inputs: LW and width, front height, and one hidden layer. The predicted model accounted for bone, with r^2^ = 0.72 and RMSE = 1.11, as shown in Table 9. The model was tested and showed an r^2^ = 0.50 and an RMSE = 1.21.

The model with LW, rump width, and chest width was the model used to estimate the amount of bone, with r^2^ = 0.05 and RMSE = 0.30 using regression tree. The best model for bone estimation was tested, and the result showed that r^2^ = 0.03, and RMSE = 0.70.

### 3.7. Summary of Results

All statistical method results were compared in terms of r^2^ and RMSE using 24 ewes at the same points of the breeding cycle, as shown in Table 10. For test data, ANN matrixes were produced to estimate dependent variables. The ANN showed the highest results for test data for fat with an r^2^ = 0.90 and an RMSE = 1.01; for muscle with an r^2^ = 0.72 and an RMSE = 1.03; and for bone with an r^2^ = 0.50 and an RMSE = 1.21.

The ANN matrixes were then used in the image-processing application to calculate body composition, as shown in Figure 4.

In summary, ANN models were the best in terms of the highest r^2^ values and lowest RMSEs for the prediction of fat, muscle, and bone, compared with MLR and RT. ANN matrixes were produced to estimate body composition using input variables.

The results of fat estimation for the test data between the ANN, MLR, and RT, along with BCS results, are shown in Table 11. The maximum difference for the MLR, compared with CT fat, was 3.1 kg, and there was a minimum difference of 0.002 kg. For ANN, there was a 2.69 kg maximum difference and a 0.034 kg minimum difference. The RT had a maximum difference of 8.34 kg and a minimum difference of 0.59 kg.

The BCS result showed a range of fat estimation between 3.52–13.03 kg for condition 2.5 and a range of 5.11–13.86 kg for a score of 3.0. Two ewes had fat amounts of 5.14 kg and 7.91 kg with condition 3.5, and another ewe had a fat amount of 4.39 kg with a condition score of 4.0. The CT fat and ANN fat results based on BCS are shown in Figure 5.

## 4. Discussion

An alternative method to predict the body composition of ewes during their production life cycle was proposed. The farmer first weighed the sheep, then took top and side images using fixed-top and side cameras where the sheep was constrained. The farmer then input the data (images and LW) to the computer, where the application was installed. The application then displayed the amount of fat, muscle, and bone. For example, if the amount of fat was less than or more than 9% of the LW, then the farmer had to evaluate the status of energy reserves and provide the required nutritional intake [40]. The setup of the new method required using top and side cameras, an electronic scale, and a little training. This allowed farmers to have their setup and use this method at any time without moving ewes away from the farm or exposing animals to machine radiation while scanning. The new method required less effort and time and did not need technical expertise and tranquillization of sheep with Acezine the way the CT method does [20]. The CT method had high r^2^ values of 0.98 for fat, 0.92 for muscle, and 0.83 for bone, whereas the new method showed lower r^2^ values of 0.90 for fat, 0.72 for muscle, and 0.50 for bone. Body fat was predicted on-farm by subjective methods, such as BCS [8]. A BCS between 2.5–3.5 meant a ewe was in good condition [2]. The development of a new BCS scale was required to provide a more accurate estimation of the fat of ewes [8]. BCS can provide a good indication of fat percentage, as stated by Tait et al. [5]. In contrast, the results of this study showed a range of fat between 1.29 kg and 13.86 kg, with an average fat of 5.55 kg for a BCS between 2.5 and 3.5. A narrower range or percentage is needed for farmers to evaluate the health of their livestock because this range is too vast. This study’s findings revealed that 9% of fat to the LW was ideal for ewes. The results showed that many ewes had the same BCS, which provided a rough indication of ewe condition, but there was a wide range of chest widths and fat measurements, confirming that the BCS was not just subjective but also inaccurate for evaluating fat. For example, one ewe had 5.2 kg of fat, a BCS of 3.0, and a chest width of 252 mm at weaning, then 7.1 kg fat, 3.0 BCS, and a chest width of 327.6 mm at pre-mating. This shows that fat and chest width increased over time, but the BCS remained the same. This is in line with other studies that showed a high variation of fat within certain BCSs [4,8]. This research used measurements such as LW and chest width, width, rump width, and front height. In the results, fat was predicted with an r^2^ = 0.90, which was higher than the study by Doeschl et al. [36], which had an r^2^ = 0.69 using rump width only, which explains that using LW will increase the accuracy. The majority of previous studies used linear methods such as MLR to predict body composition. This research used the linear method (MLR) [18,21,36,37] and non-linear methods (ANN, RT) to predict body composition and made a comparison between them. The ANN method showed the highest r^2^ values and the lowest RMSE.

## 5. Conclusions

The body composition of ewes can now be determined using a new technique that was developed using an image-processing application. Instead of relying on a wide range of BCSs between 2.5–3.5, this technique aids farmers in making decisions if the amount of fat is less than or more than 9% of the LW. The technique can be used to measure body parameter values on shorn or woolly ewes. The suggested method yields less subjective results than the BCS. This method is based on predicting body composition at two different points in the production cycle. Farmers can use the application with little training provided. Farmers can also use the equations of MLR to estimate fat with an r^2^ that reaches 0.87.

## Figures and Tables

**Figure 1 animals-13-02391-f001:**
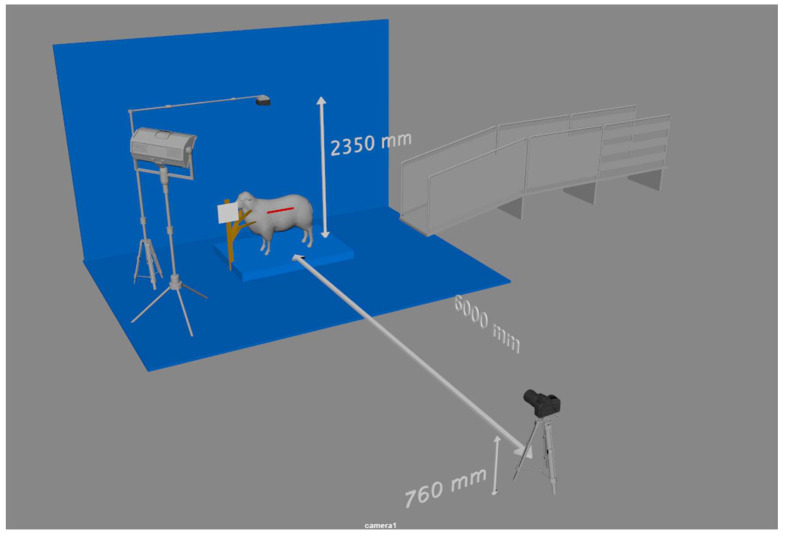
On-farm weaning and pre-mating experimental setup.

**Figure 2 animals-13-02391-f002:**
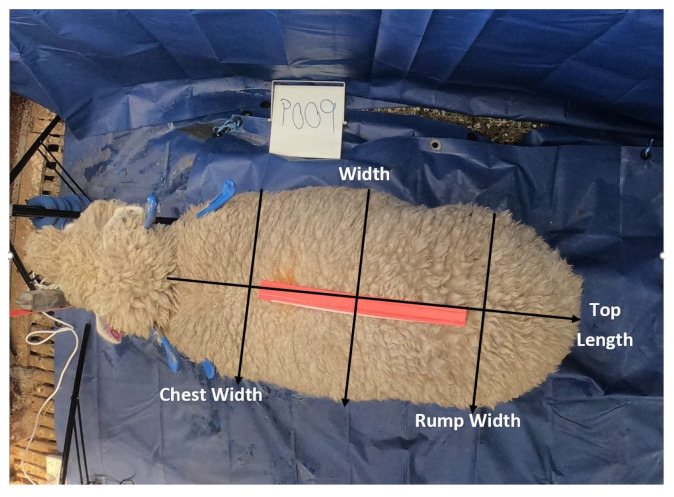
Ewe top image and body parameters.

**Figure 3 animals-13-02391-f003:**
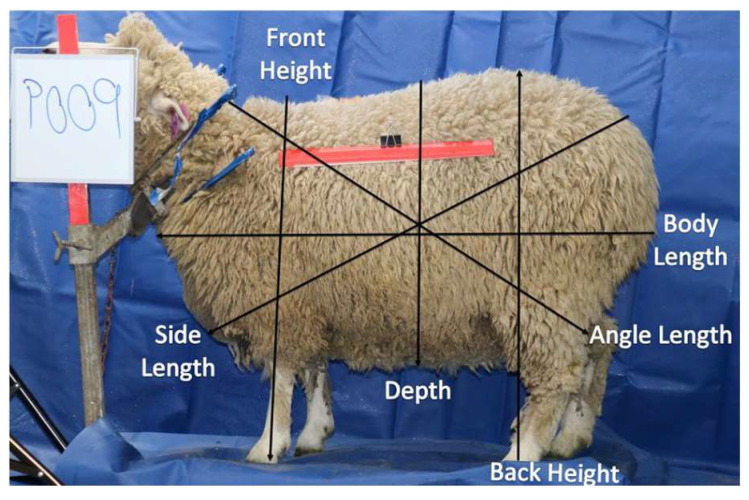
Ewe side image and body parameters.

**Figure 4 animals-13-02391-f004:**
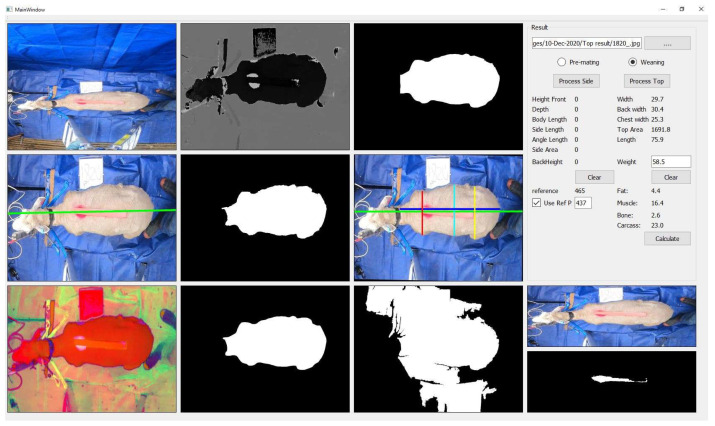
Output screen of the image-processing application.

**Figure 5 animals-13-02391-f005:**
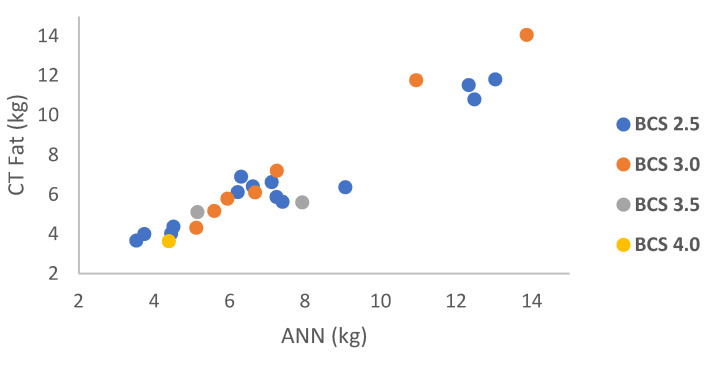
Relationship between CT fat vs. ANN fat based on BCS.

**Table 1 animals-13-02391-t001:** Descriptive statistics of the training data.

Item	Minimum	Maximum	Mean	Std Deviation
Fat (kg)	0.88	17.65	5.26	3.00
Muscle (kg)	12.65	20.78	16.22	1.49
Bone (kg)	2.03	3.77	2.68	0.32
BCS	2.0	4.5	2.72	0.52
LW (kg)	44.00	88.50	58.92	7.83
Chest width (mm)	220.3	360.2	270.2	20.8
Angle length (mm)	670.1	870.6	770.9	40.2
Body length (mm)	600.7	810.6	710.7	40.5
Side length (mm)	640.7	870.1	760.6	40.4
Front height (mm)	540.9	670.8	620.2	20.7
back height (mm)	560.1	710.6	640.0	30.0
Depth (mm)	320.6	470.0	380.2	20.6
Top length (mm)	670.5	950.7	780.6	50.4
Width (mm)	270.8	370.3	310.9	20.1
Back width (mm)	200.0	380.5	300.7	20.9
Top area (mm^2^)	13,543	288,061	197,304.6	28,909.4
Side area (mm^2^)	216,830	316,730	283,703.6	31,401.9

**Table 2 animals-13-02391-t002:** Average error % of absolute weaning and pre-mating between actual and app measurements.

Values	Angle Length	Body Length	Height	Depth	Top Length	Width	Side Length
Weaning	5%	4%	4%	3%	5%	4%	n/a
Pre-mating	7%	3%	3%	4%	6%	5%	4%

**Table 3 animals-13-02391-t003:** Rotated component matrix.

Component Matrix			
	Component		
	1	2	3
BCS		0.789	
LW		0.707	
Chest width	0.827		
Angle length	0.805		
Body length	0.834		
Side length	0.859		
Height			0.779
Back height			0.769
Depth	0.708		
Top length	0.736		
Width	0.745		
Rump width	0.726		
Top area	0.916		
Side area	0.935		

Extraction method: principal component analysis. Rotation method: varimax with Kaiser normalization.

**Table 4 animals-13-02391-t004:** Relationships between LW and body parameters to estimate fat—MLR.

Independent Variables	r^2^	Equation	RMSE
LW, Chest width	0.79	−20.043 + 0.244LW + 0.401CH	1.34
LW, Angle length	0.71	−23.159 + 0.296LW + 0.141AL	1.59
LW, Body length	0.72	−21.733 + 0.301LW + 0.129BL	1.59
LW, Side length	0.71	−21.119 + 0.293LW + 0.119SL	1.62
LW, Front height	0.68	−14.670 + 0.315LW + 0.022FH	1.69
LW, Back height	0.68	−13.195 + 0.318LW + −0.004BH	1.69
LW, Depth	0.70	−18.764 + 0.288LW + 0.185D	1.64
LW, Top length	0.69	−17.092 + 0.310LW + 0.052TL	1.67
LW, Width	0.73	−22.113 + 0.247LW + 0.402W	1.56
LW, Rump width	0.71	−17.949 + 0.303LW + 0.175RW	1.62
LW, Top area	0.73	−16.688 + 0.291LW + 0.002TA	1.55
LW, Side area	0.72	−17.402 + 0.285LW + 0.002SA	1.58
All variables	0.80	−21.115 + 0.235LW + 0.522CH + 0.101AL + −0.042BL + 0.008SL + 0.034FH + 0.013BH + −0.067D + −0.053TL + −0.021W + −0.090RW	1.34

**Table 5 animals-13-02391-t005:** Relationships between LW and body parameters to estimate fat—ANN.

Independent Variables	r^2^	RMSE
LW, Chest width	0.88	1.17
LW, Angle length	0.84	1.61
LW, Body length	0.83	1.36
LW, Side length	0.82	1.33
LW, Front height	0.80	1.81
LW, Back height	0.81	1.78
LW, Depth	0.85	1.47
LW, Top length	0.85	1.44
LW, Width	0.84	2.18
LW, Rump width	0.84	2.21
LW, Top area	0.84	2.21
LW, Side area	0.85	1.28
All variables	0.95	1.22

**Table 6 animals-13-02391-t006:** Relationships between LW and body parameters to estimate muscle—MLR.

Independent Variables	r^2^	Equation	RMSE
LW, Chest width	0.51	10.773 + 0.156LW + −0.138CH	1.04
LW, Angle length	0.47	9.564 + 0.134LW + −0.015AL	1.09
LW, Body length	0.47	9.700 + 0.134LW + −0.019BL	1.09
LW, Side length	0.47	9.778 + 0.135LW + −0.045SL	1.09
LW, Front height	0.47	5.919 + 0.127LW + 0.045FH	1.08
LW, Back height	0.47	7.476 + 0.129LW + 0.018BH	1.09
LW, Depth	0.47	10.112 + 0.140LW + −0.056D	1.08
LW, Top length	0.47	8.345 + 0.131LW + 0.002TL	1.09
LW, Width	0.52	12.588 + 0.164LW + −0.189W	1.03
LW, Rump width	0.48	10.146 + 0.137LW + −0.064BW	1.07
LW, Top area	0.49	9.500 + 0.139LW + −0.001TA	1.07
LW, Side area	0.47	9.303 + 0.138LW + 0.000SA	1.58
All variables	0.52	−5.109 + 0.151LW + −0.082CH + −0.004AL + 0.020BL + −0.044SL + 0.010FH + −0.004BH + −0.046D + −0.102TL + −0.004W + 0.072RW + −0.003TA + 0.001SA	1.4

**Table 7 animals-13-02391-t007:** Relationships between LW and body parameters to estimate muscle—ANN.

Independent Variables	r^2^	RMSE
LW, Chest width	0.76	1.13
LW, Angle length	0.63	1.87
LW, Body length	0.62	1.01
LW, Side length	0.73	1.11
LW, Front height	0.74	1.03
LW, Back height	0.73	1.33
LW, Depth	0.71	2.42
LW, Top length	0.63	1.09
LW, Width	0.65	1.11
LW, Rump width	0.66	1.07
LW, Top area	0.71	1.01
LW, Side area	0.72	1.09
All variables	0.79	1.20
LW, Rump width, Front height	0.77	1.26

**Table 8 animals-13-02391-t008:** Relationships between LW and body parameters to estimate bone—MLR.

Independent Variables	r^2^	Equation	RMSE
LW, Chest width	0.22	1.616 + 0.021LW + −0.006CH	0.89
LW, Angle length	0.24	0.861 + 0.018LW + 0.010AL	0.88
LW, Body length	0.24	0.952 + 0.019LW + 0.009BL	0.88
LW, Side length	0.24	0.947 + 0.018LW + 0.009SL	0.88
LW, Front height	0.23	1.133 + 0.019LW + 0.007FH	0.89
LW, Back height	0.22	1.317 + 0.019LW + 0.004BH	0.89
LW, Depth	0.25	2.152 + 0.023LW + −0.022D	0.88
LW, Top length	0.25	0.808 + 0.018LW + 0.010TL	0.88
LW, Width	0.26	2.321 + 0.026LW + −0.037W	0.87
LW, Rump width	0.22	1.553 + 0.020LW + −0.001BW	0.89
LW, Top area	0.22	1.474 + 0.019LW + −0.005TA	0.89
LW, Side area	0.22	1.458 + 0.019LW + 0.005SA	0.89
All variables	0.36	1.678 + 0.029LW + −0.035CH + −0.008AL + 0.0006BL + −0.013SL + −0.017FH + 0.019BH + −0.063D + −0.026TL + −0.067W + 0.023RW + −0.001TA + 0.000SA	0.25

**Table 9 animals-13-02391-t009:** Relationships between LW and body parameters to estimate bone—ANN.

Independent Variables	r^2^	RMSE
LW, Chest width	0.45	2.3
LW, Angle length	0.50	2.31
LW, Body length	0.43	1.03
LW, Side length	0.41	1.05
LW, Front height	0.60	1.82
LW, Back height	0.42	1.04
LW, Depth	0.57	1.94
LW, Top length	0.65	1.15
LW, Width	0.65	1.05
LW, Rump width	0.50	1.03
LW, Top area	0.58	1.01
LW, Side area	0.56	2.22
All variables	0.75	2.40
LW, Chest width, Front height	0.59	1.2
LW, Angle length, Front height	0.46	1.12
LW, Body length, Front height	0.53	1.19
LW, Side length, Front height	0.52	2.17
LW, Depth, Front height	0.61	1.0
LW, Top length, Front height	0.53	2.36
LW, Width, Front height	0.72	1.11

**Table 10 animals-13-02391-t010:** Test data prediction of 24 ewes—ANN vs. MLR vs. RT.

Dependent Variables	Independent Variables		
	MLR − r^2^	ANN − r^2^	RT − r^2^
Fat	0.87 (LW, chest width)	0.90 (LW, chest width)	0.74 (LW, chest width)
Muscle	0.41 (LW and width)	0.72 (LW, rump width, front height)	0.21 (LW, width and chest width)
Bone	0.34 (LW and width)	0.50 (LW, width, front height)	0.03 (LW, rump width and chest width)

**Table 11 animals-13-02391-t011:** Fat test data estimation in (kg)—body condition score.

CT Fat	MLR	ANN	RT	BCS
13.867	11.240	14.062	6.92	3
12.479	9.588	10.800	6.5	2.5
7.249	7.680	7.196	5.93	3
4.390	3.631	3.639	11.51	4
6.674	6.035	6.105	8.59	3
9.066	7.157	6.372	3.5	2.5
6.209	6.974	6.125	5.53	2.5
7.919	6.230	5.608	14.12	3.5
12.329	9.850	11.521	3.98	2.5
7.242	6.364	5.874	3.83	2.5
5.117	4.656	4.324	4.05	3
5.943	6.084	5.790	8.99	3
13.034	9.876	11.809	6.1	2.5
4.451	4.185	4.036	6.63	2.5
3.742	4.141	4.007	8.27	2.5
4.513	4.518	4.379	5.16	2.5
10.940	10.129	11.774	6.62	3
7.403	6.486	5.627	6.66	2.5
6.301	7.687	6.893	5.71	2.5
5.148	5.177	5.114	10.34	3.5
5.595	5.771	5.176	8.41	3
7.114	7.567	6.620	8.12	2.5
6.616	7.314	6.411	2.67	2.5
3.525	3.653	3.666	7.687	2.5

## Data Availability

The data presented in this study are available upon request from the corresponding author.

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
