# Peer review of "Body Composition Estimation in Breeding Ewes Using Live Weight and Body Parameters Utilizing Image Analysis"

_animals, 2023, doi:10.3390/ani13142391_

Round 1
Reviewer 1 Report
The manuscript develops a tool to determine the body composition of live ewes using a non-destructive method. Although the results are not as good as resonance, the alternative method offers important advantages for its real use on farms. This topic is relevant in practice because simple and non-destructive tools to assess the body composition of livestock are always useful and interesting.
As a novelty, the authors have compared resonance with measurements that are easy to develop in farming conditions, filling a gap in this area of knowledge. As an improvement to go further, the authors could dissect carcasses to validate the resonance data. The title, results and conclusions are consistent and the results presented address the aim of the manuscript. Authors have used current references and there are no unjustified self-references.
As the introduction of the manuscript is long, it could be shortened and include a statement explaining why MLR and ANN or other tools were used.
Minor questions:
Image captions should be more explicative.
There are some missing references, mainly in material and methods section.
Just a final question in L304. Are the same ewes?
Author Response
Hi There
Thank you for the comments, I really appreciate it!
Regards
Ahmad

Reviewer 2 Report
The manuscript “Body Composition Estimation in Breeding Ewes Using Live Weight and Body Parameters Utilizing Image Analysis” gives an interesting approach to decrease the subjectivity of the body condition score measurements by providing an alternative for body composition estimation in ewes. Nonetheless, the presentation of the manuscript is poor; my recommendation is to reconsider after major corrections because of the following flaws:
The manuscript needs to be rewritten; it is difficult to understand, some paragraphs are repetitive, and many references are missing, showing the legend “Error! Reference source not found..”. This is unacceptable. Also, active references are incorrectly presented (L605, 634, among others).
The introduction is too long and repetitive; some materials and methods are mixed(L73-81).
The materials and methods section is confusing, have results mixed (309-311), and some paragraphs are repeated (314-321 and 383-394).
Results and discussion are interesting, but it is recommended to be more concise. Be careful with the references.
Another recommendation is to be more precise on the BCS measuring method used, as there are some differences between them (1-4 scale, 1-5 scale, full point increases, 0.5 increases, etc.).
English language has some minor mistakes and needs reviewing, but it is understandable. the main issue is with the writing style.
Author Response
Hi
Thank you for the comments, I really appreciate it!
Regards
Ahmad

Reviewer 3 Report
Review to MS ID animals-2461701
Body condition scoring is a highly practical method to predict actual energy/reserve status of ewes in special production stages. However it is a very subjective determination of the animals’ condition and its’ accuracy and repeatability depend on several factors like inspector, seasons, ewe’s production stages, age, breeds etc.
Authors would like to develope a more profound individual prediction method for body composition of ewes related to body measurements parameters.
„This research investigates the use of body parameters determined by image processing and LW to estimate body composition during ewe production cycle instead of using BCS (indicates fat only).”
Title is adequate,the background of the research is widely assumed in the introduction section. Many influencing factors of BCS process, body compositions and body measurements are described.
In case of materials and methods a few finetuning are needed to improve this part of the manuscript. Several repetition were found in this section, duplication should be avoided. Furthermore exact manufacturer names of used equipments should be given in case of special measurements or processes. There are no data about investigated breed, breeding protocol, feeding and keeping management. It is illustrated well.
Results are clear and all the data were summarized in 12 tables and and visualized in one figure (this last one is very tricky/interesting). What was the age composition in the investigated ewes group? Alltogether the collected data (BW, BCS, body measurements ) were validated by CT measurement. The highest R2 values were 0.90, 0.70 and 0.51 for fat, lean and bone with ANNs respectively. The most informative parameters were LW and chest width for prediction of fat contents. The newly developed method could be used for shorn ewes or ewes in wool.
Article is recommended to publish after improvement of the methodological description of the trial.
Author Response

(The authors gave the same response as above.)

Round 2
Reviewer 2 Report
There is still a big issue with the manuscript, the information obtained is interesting, but the statements seem a little ambitious; I don’t see the method being used instead of the BCS measurements in-farm. Even if it is more accurate, it needs more specialized equipment or training, either pictures or rulers or apps. I would be more conservative with the used statements (i.e. L1788-9 “An objective on-farm method to predict the body composition of ewes during their production life cycle was established”. L1965 “The established method”).
In summary, the research has scientific merit, but it still needs to be rewritten to correctly transmit its findings and importance.
L58: change “and” to “of”
L59: “information to inform” sounds redundant consider rewriting
Consider changing the term “lean”; I understand that it refers to meat with a low amount of fat, maybe use “muscle” instead, or define “lean” (less than 10 g of fat in 100 g of meat?).
L185: change to “breed”
The introduction is still very long and needs to be more concise; just justify why the body composition estimation is necessary and why it would be useful to farmers. The literature revision is unnecessary.
Even though the materials and methods have been improved, I get lost in what the actual on-farm method is. Is it a set of pictures, a ruler, or an app? Be clearer about the method; anybody should be able to recreate it.
The English language has minor mistakes, and the writing style still needs work, but it has been improved.
Author Response
Hi Reviewer
Again, thank you for your insightful remarks and feedback.
To demonstrate the significance and findings of my research, I revised the paper and included further information.
With regards
Ahmad

Round 3
Reviewer 2 Report
All the proposed corrections have been made, and manuscript is much more comprehensible. Therefore, I believe it is ready for publishing.